# Trace elements concentration in blood of nesting Kemp's Ridley turtles (*Lepidochelys kempii*) at Rancho Nuevo sanctuary, Tamaulipas, Mexico

**Kevin Alan Zavala-Félix**[1], **Miguel Angel Reyes-López**[2], **Fátima Yedith Camacho-Sánchez**[2], **Héctor Hugo Acosta-Sánchez**[3], **Catherine E. Hart**[4], **Alan A. Zavala-Norzagaray**[1], **Valeria Leal-Sepúlveda**[1], **Renato Leal-Moreno**[1], **Brenda Aracely Espinoza-Romo**[1], **A. Alonso Aguirre**[5], **César P. Ley-Quiñónez**[1]*

1 Laboratory Vida Silvestre, CIIDIR Sinaloa- Department Medio Ambiente, Instituto Politécnico Nacional, Guasave, Sinaloa, Mexico, 2 Centro de Biotecnología Genómica-Conservation Medicine Laboratory, Instituto Politécnico Nacional, Reynosa, Tamaulipas, Mexico, 3 Programa de Conservación de Tortugas Marinas en el Santuario Playa de Rancho Nuevo, Terra Asesoría Ambiental S.C., Ciudad Victoria, Mexico, 4 Investigación, Capacitación y Soluciones Ambientales y Sociales AC, Tepic, Nayarit, México, 5 Warner College of Natural Resources, Michael Smith Natural Resources Building, Colorado State University, Fort Collins, CO, United States of America

* cleyq@ipn.mx

## Abstract

The concentrations of trace elements including As, Zn, Cu, Se, Pb, Hg and Cd, were determined in the blood of nesting Kemp's ridley turtles (*Lepidochelys kempii*) at Rancho Nuevo sanctuary, Tamaulipas, Mexico during 2018–2020. The sequential concentrations analyzed were Zn> Se> Cu> As> Pb; while Cd and Hg concentrations were below the limits of detection (0.01 µg g$^{-1}$). No significant differences were observed between the concentrations of trace elements (*p> 0.05*) by year, except Se levels, possibly resulting from recorded seasonal differences in turtle size. No relationships among turtle size *vs* elements concentration were observed. In conclusion, essential and toxic trace elements concentrations in the blood of nesting Kemp's ridley turtles may be a reflex of the ecosystem in which the turtles develop, that is, with low bioavailability of elements observed in the trophic webs in the Gulf of Mexico.

## Introduction

Coastal habitats are negatively impacted by waste produced through agriculture, mining, urbanization, fisheries, and the oil industry. These waste products are released into the environment increasing contamination levels [1–4] which affect the health of species and ecosystems [5,6]. Semi-enclosed seas are particularly affected where anthropogenic activities increase the bioavailability of trace elements. Due to their speciation capacity, trace elements are persistent in the environment [7]. Therefore, organisms are under continuous stress due to contamination [8–10]. Pollution levels increase through bioconcentration, bioaccumulation and biomagnification along the trophic web, affecting organisms such as sea turtles further up

**Data Availability Statement:** The Data is available in the next repository: https://doi.org/10.7910/

DVN/W56DLC Ley-Quiñónez, Cesar P., 2022, "Trace elements concentration in blood of nesting Kemp's Ridley turtles (Lepidochelys kempii) at Rancho Nuevo sanctuary, Tamaulipas, Mexico.", https://doi.org/10.7910/DVN/W56DLC, Harvard Dataverse, DRAFT VERSION.

**Funding:** The author(s) received no specific funding for this work.

**Competing interests:** The authors have declared that no competing interests exist.

these networks altering their metabolic pathways and increasing the potential for disease and death [11–16].

Kemp's ridley turtles (*Lepidochelys kempii*) are considered the most critically endangered of all sea turtle species by the IUCN [17,18]. It is endemic to the Gulf of Mexico with 90% of the population nesting in Rancho Nuevo Sanctuary, Tamaulipas, Mexico [19,20]. Kemp's ridleys face multiple threats induced by environmental contamination present in the Gulf of Mexico caused by hydrocarbons, organochlorine compounds, carbamates, solid waste, pharmaceuticals, macro and microplastics and toxic trace elements [21–25]. The latter is of particular concern due to the dominant anthropogenic activities in the region [26,27]. These include fertilizer production, mining, and oil refining.

The oil industry is the principal contributor due to the large amounts of crude oil and waste products that have spilled into coastal areas over the years, apporting certain elements like Cd, Zn, Cu, Pb, As, Hg, etc. [28–30]. In 2010, the Gulf of Mexico was affected by the Deepwater Horizon oil spill [31], affecting Kemp's ridley foraging areas [32]. The incident impacted over 61,000 Kemp's ridley turtles that stranded directly or indirectly linked to this event and representing approximately 35% of a total estimated population of almost 178,000 in 2013. Current trends demonstrate that the species is recovering with recruits arriving to nesting beaches annually [33]. In addition, toxic elements remain a potential threat to Kemp's ridley turtles [31,34–36]. The present study aimed to quantify the concentrations of trace elements in blood of nesting Kemp's ridley turtles at Rancho Nuevo Sanctuary, Tamaulipas, Mexico. This information may be useful to provide a better understanding of bioaccumulation process and possible population health impacts on this endangered species.

## Materials and methods

### Sample collection

Blood samples were collected from nesting Kemp's ridley turtles at the Rancho Nuevo Sanctuary, Tamaulipas, Mexico (23˚10'54" N, - 97˚46'05" W) during the mass nesting arribada seasons occurring April to July 2018 to 2020. Blood was collected from the dorsal cervical sinus according to previous studies [37]. Briefly, once the turtle had finished ovipositing, the blood sample was collected by tilting the individual at a minimum angle of 30˚, supported by a mound of sand, and the neck was slightly stretched to increase blood flow to the anatomical region [38]. A total of 5 mL of blood was collected with 21Gx½ gauge double-ended syringe and needle and stored in 10 mL tubes with ethylenediaminetetraacetic acid (EDTA) as anticoagulant (Beckton-Dickinson, Franklin Lakes, NJ). The samples were refrigerated at 4˚C until laboratory processing [37].

### Female biometrics and tagging

For each turtle, curved carapace length (CCL) notch to tip, straight carapace length (SCL) and curved carapace width (CCW) [39] were using calipers and a flexible measuring tape [45]. Each turtle was tagged on the second scale of their left flipper with one Inconel tag, and one intradermal passive integrated transponder (PIT) tag when available, in order. To record recaptures, each turtle underwent a visual examination and was assigned to the category best describing its general physical condition as: healthy or injured [40]. Body condition was established based on the concavity of the plastron [41] where a concave plastron indicated poor health, a flat plastron denoted a fair condition, and a convex shape reflected good health. The quantity and size of fibropapillomas were evaluated following the method by Work and Balazs [42] and epibiont load was categorized using a scale of 1 to 3 with 1 = mild: <20 epibionts; 2 = moderate: 20–50 epibionts; and 3 = high: > 50 epibionts [43].

## Trace elements analysis

Trace elements analyzed included Zn, Cu, Se, Hg, Pb, Cd and As. Acid digestions of the blood samples obtained were performed for their determination using methodology previously described [14]. An acid mixture of 5 mL of $HNO_3$ and HCl in a 4:1 ratio was added to 0.5 g of whole blood from each sample, using a microwave digestion system (MARS Xpress CEM). Each digestion was measured with deionized water in 25 mL volumetric polypropylene flask and refrigerated until analysis, which occurred in a period not exceeding 48 h after digestion to avoid volatilization or adsorption by the flask walls. Toxic and essential trace elements concentration analysis was performed using Inductively Coupled Plasma Optical Emission Spectroscopy (ICP-OES, VARIAN 730-ES). The detection limits of the equipment were 0.5 mg $kg^{-1}$ for Hg and 0.02 μg $g^{-1}$ for all other elements analyzed.

Reference materials certified by the National Research Council of Canada (TORT-3) were used as quality controls and to determine the percentage of evaporation and recovery of the analyzed trace elements. Analyzes were performed in duplicate fortified with standards of reference (Perkin Elmer GFAAS Mixed Standard). Blanks (deionized water) were placed every eight samples and underwent the same digestion process to detect possible contamination [34,44]. The final digestions were clear and transparent; likewise, the recovery percentage of the analyzed trace elements was between 89–106%.

## Statistical analysis

Data normality was assessed by the Kolmogorov Smirnov normality test. Statistical data were reported as arithmetic means ± standard error (mean ± SE) and range (minimum-maximum). Trace elements concentrations were presented in micrograms per gram wet weight (μg $g^{-1}$). The one-way analysis of variance (ANOVA) parametric test ($\alpha = 0.05$) and Tukey's multiple comparison test were used to assess differences regarding elements concentrations and individual biometry data. The Kruskal-Wallis test was used to analyze non-parametric data. A simple regression model ($R^2 > 50\%$) was performed to find the statistical relationship between the trace elements concentrations and the biometrics.

## Ethics statement

Permits were granted in Mexico by Dirección General de Vida Silvestre/Secretaría para el Medio Ambiente y los Recursos Naturales (SEMARNAT) to study and manage wildlife samples or species. Permit numbers: SGPA/DGVS/04674/10 and SGPA/DGVS/003769/18.

## Results and discussion

During the 2018 to 2020 nesting seasons, 83 blood samples were collected from nesting Kemp's ridley turtles at Rancho Nuevo beach, Tamaulipas, Mexico. All turtles captured were in good health, without wounds or external fibropapillomas and presented low or no epibiotic load. The average nesting female size was SCL of 60.66 ± 0.28 cm and a CCL of 65.315 ± 0.34 cm (Table 1). Turtles measured in 2020 were significantly smaller (SCL: 59.46 ± 0.33) than in

**Table 1. Morphometric data (cm) of *L. kempii* turtles from Rancho Nuevo, Tamaulipas, Mexico, 2018–2020.**

|       | Mean±SE       | (min-max)       |
|-------|---------------|-----------------|
| SCL   | 60.66±0.28    | (55.74–65.88)   |
| CCL   | 65.315±0.34   | (59.20–71.80)   |
| CCW   | 64.57±0.46    | (56.60–72.60)   |

SCL = Straight Carapace Length. CCL = Curved Carapace Length. CCW = Curved Carapace Width.

**Table 2. Heavy metal concentrations reported in different areas (mean ± standar deviation, µg g$^{-1}$ wet weight) in blood of Kemp´s ridley turtles.**

| Area | Nesting (*This study*) | Nesting (Wang, 2005) | Foraging (Orvik, 1997) | Foraging (Wang, 2005) | Foraging (Wang, 2005) | Foraging* (Perrault et al., 2017) |
|---|---|---|---|---|---|---|
| As | 0.08±0.03 | NA | NA | NA | NA | 6.84±1.98d |
| Hg | ND | 0.06±0.04 | 0.018 (0.0005–0.06) | 0.01±0.009 | 0.01±0.01 | 0.04±0.04d |
| Cd | ND | 0.01±0.01 | NA | 0.007±.005 | 0.01±0.005 | 0.02±0.01d |
| Cu | 0.09±0.01 | 0.40±0.09 | 0.52 (0.21–1.3) | 0.47±0.06 | 0.41±0.11 | NA |
| Pb | 0.06±0.02 | 0.05±0.02 | 0.001 (0.00–0.03) | 0.02±0.03 | 0.03±0.03 | 0.01±0.004d |
| Se | 0.14±0.05 | NA | NA | NA | NA | 4.11±1.83d |
| Zn | 0.79±0.79 | 22.70±12.6 | 7.5(3.28–18.9) | 3.9±1.47 | 6.71±4.46 | NA |

* = Analysis performed in red blood cells. **NA** = Not analyzed. **ND** = Not detected. In parentheses min-max when no standar deviation is reported.

2018 (SCL: 62.77 cm ± 0.52) and 2019 (SCL: 61.88 cm ± 0.33) (*p <0.05*). The turtles in this study were young females possibly laying their first clutch [45]. Kemp's ridley turtles become sexually mature between 8 and 12 years of age with a first clutch laid at an average size of SCL 61.8 ± 1.8 cm [19,20,45,46]. This is encouraging as new nesting females recruiting to this important rookery are contributing to the species recovery [45]. This coincides with Caillouet Jr [33] who found Kemp's ridley recruits in neritic areas and nesting beaches, corresponding to the age of maturation and nesting of turtles hatching after 2010. As previously stated, the Deepwater Horizon oil spill in the northeast of the Gulf of Mexico occurred during 2010 and directly or indirectly impacting 34.5% of the Kemp's ridleys population [32], this suggests that there are young adults in the nesting population that have not bioaccumulated high concentrations of toxic elements. The blood analysis documented that essential elements were more abundant compared to toxic ones, with a distribution Zn> Se> Cu> As> Pb. The concentrations of Hg and Cd were below detection limits (Table 2). No significant differences were observed between the concentrations of trace elements (*p> 0.05*) by year, except for Se, where concentrations were higher in 2018 than those found in 2020, *p = 0.035* (Table 3). Similarly, Pb and Cu concentrations of 2020 samples were below detection limits.

Currently, work is underway to establish basal values of trace elements concentrations in nesting Kemp's ridley blood. Their bioavailability and bioaccumulation in sea turtles are influenced by multiple factors including species, life stage, diet, individual condition, climatic factors, and region [14,47–51]. Perhaps, feeding represent the main source of trace elements found in sea turtles [52]. The trophic position of the species plays a key role in bioaccumulation and biomagnification processes [14,53–55].

**Table 3. Heavy metal concentrations (mean ± standar deviation, µg g$^{-1}$ wet weight) in blood of nesting Kemp´s ridleys (*Lepidochelys kempii*) from Rancho Nuevo, Mexico, 2018–2020.**

| Metal | 2018 | 2019 | 2020 | Statistical test |
|---|---|---|---|---|
| Zn | 1.02±0.17 (0.09–2.37) | 0.70±0.14 (0.10–2.14) | 0.67±0.13 (0.10–2.27) | *p = 0.207* |
| Cu | 0.09±0.002 (26) (0.07–0.11) | 0.09±0.002 (28) (0.06–0.11) | ND | *p = 0.523* |
| Pb | 0.06±0.005 (21) (0.02–0.11) | 0.06±0.003 (26) (0.03–0.10) | ND | *p = 0.339* |
| As | 0.09±0.007 (24) (0.04–0.16) | 0.08±0.003 (23) (0.05–0.11) | 0.07±0.004 (24) (0.04–0.12) | *p = 0.193* |
| Se | 0.17±0.02[a] (8) (0.08–0.25) | 0.15±0.01[ab] (11) (0.06–0.21) | 0.12±0.005[b] (18) (0.08–0.16) | *p = 0.035* |
| Cd | ND | ND | ND | NA |
| Hg | ND | ND | ND | NA |

ND = Not detected; NA = Not analyzed

n[a] = Number of samples above the detection limit. Letters indicate significant difference between groups. Statistical test: ANOVA.

The diet of Kemp's ridley turtles varies depending on their life stage. Blue crabs (*Callinectes sapidus*) are the principal food of adult kemps ridleys, while juveniles feed mainly on tunicates around nearshore islands. During the post hatchling pelagic stage little is known about their diet [56–58]. These changes in diet, may result in varying levels of trace elements in Kemp's ridleys throughout their life.

Zn was the most common element in organisms of the essential elements analyzed, plays a vital function in the growth and development and acts as a detoxifier [59], by induction metallothioneins [52,60]. However, high Zn levels can be toxic [7] and deficiencies in nesting turtles can decrease the number of eggs laid, and result in hatchling deformities [55]. This element occurs in higher levels in green turtles (*Chelonia mydas*), due to their herbivorous diet as adults, which includes algae that bioaccumulate Zn [44].

The Zn concentrations found in this study were lower (0.79 ± 0.08 µg g-1) than those previously reported for this population [61,62]. It has been mentioned that Zn concentrations are also dependent on age and size, with larger turtles accumulating higher concentrations of this element [62]. Cu is essential for growth and development even at low concentrations [48,59,63]. During vitellogenesis, both Cu and Zn concentrations decrease in nesting turtles due to the vertical transfer from the female to her eggs [63]. In addition, turtles present little or no feeding during nesting, reducing potential bioaccumulation during this period [48,51]. However, turtles nest two to three times per season [20,64], so essential elements concentrations may decrease over the nesting season [48].

Se is another essential element for sea turtles [48,65], which has antioxidant, immunological and thyroid functions [66]. Previously, a positive relationship between Hg and Se has been identified, as Se participates in the Hg detoxification processes in organisms. This correlation has not been previously reported in Kemp's ridley turtles, possibly as a result of the low levels of Hg in the population documented herein [57]. Although high concentrations can be toxic and cause neurological and dermal damage and decreased sea turtle hatching success [14,66,67]. The concentrations identified in this study were lower than those reported in other species of sea turtles worldwide [14,34,68,69].

Previous studies have shown that the distribution of essential and toxic elements in sea turtle blood presents higher levels of essential elements than toxic ones [62,69,70]. This distribution may be affected when intoxication or pathological responses occur; for example, a study in Brazil reported higher concentrations of Pb compared to Zn and Cu in *C. mydas* when these turtles presented fibropapillomatosis [50].

This study identified similar Pb levels to those previously reported (0.05 µg g-1) in Kemp's ridley turtles [62]. Despite the occurrence of the largest oil disaster in the Gulf of Mexico in 2010 [71], there has been no variation in blood Pb levels (Table 2) in the Kemp's ridley nesting turtles analyzed. However, Pb contamination has been present in the marine environment as a result of leaded gasoline, which through combustion, releases Pb into the environment and transported through biogeochemical cycles to the oceans. Most likely, Pb levels have decreased since policy change to unleaded gasoline [52,72,73]. However, it is important to continue monitoring Pb levels as this highly toxic metal can affect the nervous system and fetus development, cause infertility, immunosuppression, and osteoporosis due to its mimicry to Ca [7,35,72,74]. It has been considered that a low concentration of Pb in sea turtles should be less than 0.5 µg g$^{-1}$ [75], therefore, the levels of Pb found in the nesting Kemp's ridley turtles in this study can be considered normal for the species. These acceptable levels are consistent with those reported in nesting olive ridley turtles (*Lepidochelys olivacea*) at 0.19 ± 0.03 µg g$^{-1}$ in the Mexican Pacific [72].

Cd is considered one of the toxic metals with the highest impact and importance in ecotoxicology [7]. Cd can cause kidney, neurological and bone damage, is carcinogenic and

teratogenic even at low levels in sea turtles [15,76]. Furthermore, maternal transfer of Cd to turtle eggs occurs through vitellogenin and proteins similar to Se (selenoproteins), a process that happens in competition with other essential elements [48,59,60,76]. Species such as loggerhead (*Caretta caretta*), green, and olive turtles present higher loads of Cd due to their diet. For example, green turtles feed on algae that bioaccumulate Cd, while other turtle diets include cephalopods which introduce Cd to their diet [34,52,77]. Blood Cd levels in Kemp's ridleys were below detection limits due to the low bioavailability; consistent with those reported by previous studies (0.007 to 0.02 μg g$^{-1}$) for both juveniles and adults [35,61,62].

As is a toxic element that frequently occurs in low concentrations in sea turtles [2]. Although it occurs mainly in organic form, which is less toxic, the inorganic fraction of this element (2–10%) can be toxic to sea turtles [2,77] and may generate immune responses such as oxidative stress [35] and possible liver and kidney damage [78]. As has only been reported in one previous study in juvenile Kemp's ridleys foraging in Florida, USA [35]. The study reported higher levels of As than those found in this present study (Table 2). This is possibly related to the diet of the juvenile turtles which consists principally of tunicates which are bioacummulators of As [79] as compared to adult Kemp's ridley diet based on crustaceans [35,56,80].

Hg concentrations obtained in the present study were below detection limits (<0.5 mg kg$^{-1}$). Previous studies [57] reported a Hg concentration of 0.024 μg g$^{-1}$ in juvenile Kemp's ridley turtles highlighting that Hg vertical transmission has not been observed during vitellogenesis in this species. Most likely, exposure to this toxic element may occur during the pelagic stages, and during growth, Hg levels decrease through excretion. Hg can present pathologies in sea turtles, even in low concentrations of 0.009 μg g$^{-1}$, may cause immunosuppression [53,54,78], and be a cofactor in the development of fibropapillomas [35].

Trace elements levels in water and organisms such as fish, red crabs (*Chaceon quinquedens*) and blue crabs (*Callinectes sapidus*), are low, particularly Cd and Hg, since these are not bioavailable in the water column or sea turtle prey in the Gulf of Mexico [81,82]. Sediments their present low concentrations of Cd and Hg, whereas Zn and Pb may be found at higher levels. Interestingly, these elements remain trapped in the sediments and are not bioavailable for organisms, including benthivorous species [81].

Statistically significant relationships have been observed among Cd, Pb and As *vs* Zn and Cu, since these two essential elements can act as detoxifiers of toxic elements [59], through the induction of metallothioneins in sea turtles [52,60]. In addition, Se plays an important role as an antagonist and detoxifier of toxic elements such as Hg [6,34,66]. A positive relationship between trace elements concentrations and turtle life stage has been observed [77,83]. However, in the present study no relationships were identified, neither between elements analyzed nor between turtle size *vs* trace elements concentration ($R^2$<50%). Similar results were reported previously for juvenile Kemp's ridley turtles [41]; therefore, bioconcentration is not associated with age unlike other sea turtle species.

## Conclusions

Kemp's ridley turtles demonstrated low levels for most trace elements analyzed in their blood. This may be reflective of the ecosystem in which the turtles develop, that is, with low bioavailability of trace elements observed in the trophic webs in the Gulf of Mexico. The low levels of these contaminants present in the potential prey of Kemp's ridley turtles, most likely do not represent a risk to the health of this nesting population. However, some toxic trace elements such as Hg can present speciations such as methylmercury, that at low concentrations, produce sublethal toxicity at the cellular level and immunosuppression. Currently, there are no

maximum permissible limits of trace metals for sea turtles and no published blood reference values for Kemp's ridley turtles. Therefore, it is difficult to establish the concentration at which sea turtle health is at risk, particularly for metals such as Cd and Hg. Further research is needed on the speciation of some metals like mercury and the possible health impacts on endangered Kemp's ridley turtles and should consider using equipment with greater precision to study the low levels of Cd and Hg found in this study, as these metals are important in ecotoxicology.

## Acknowledgments

We would like to thank the community of Rancho Nuevo Tamaulipas, Mexico and especially Juan Martínez for his support and to Martha López, director of CONANP Tamaulipas.

## Author Contributions

**Conceptualization:** Kevin Alan Zavala-Félix, Miguel Angel Reyes-López, Catherine E. Hart, Renato Leal-Moreno, César P. Ley-Quiñónez.

**Data curation:** Kevin Alan Zavala-Félix, Fátima Yedith Camacho-Sánchez, Valeria Leal-Sepúlveda, Brenda Aracely Espinoza-Romo.

**Formal analysis:** Kevin Alan Zavala-Félix, Miguel Angel Reyes-López, Catherine E. Hart, Alan A. Zavala-Norzagaray.

**Investigation:** Kevin Alan Zavala-Félix, Miguel Angel Reyes-López, Héctor Hugo Acosta-Sánchez, César P. Ley-Quiñónez.

**Methodology:** Kevin Alan Zavala-Félix, Miguel Angel Reyes-López, Fátima Yedith Camacho-Sánchez, Héctor Hugo Acosta-Sánchez, Alan A. Zavala-Norzagaray, Valeria Leal-Sepúlveda, Renato Leal-Moreno, Brenda Aracely Espinoza-Romo, César P. Ley-Quiñónez.

**Resources:** Miguel Angel Reyes-López, César P. Ley-Quiñónez.

**Supervision:** Héctor Hugo Acosta-Sánchez, A. Alonso Aguirre.

**Validation:** A. Alonso Aguirre.

**Writing – original draft:** Kevin Alan Zavala-Félix, César P. Ley-Quiñónez.

**Writing – review & editing:** Kevin Alan Zavala-Félix, Miguel Angel Reyes-López, Catherine E. Hart, A. Alonso Aguirre, César P. Ley-Quiñónez.

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
