## [Decision Letter · Decision Letter 0]

22 Jun 2022

PONE-D-22-12917Determination of heavy metals in blood of nesting Kemp’s Ridley turtles ( Lepidochelys kempii ) at Rancho Nuevo sanctuary, Tamaulipas, Mexico.PLOS ONE

Dear Dr. Ley-Quinonez,

Thank you for submitting your manuscript to PLOS ONE. I have now received two reviews, and you will see the reviewers differ in their ultimate judgement, but there are some common themes. Both reviewers would like to see more information on elemental recoveries, quality assurance, and at least one of the reviewers presents concerns about the detection limits for the trace elements analyzed and methodology used. I think it would be most fair to allow you to respond to these comments and revise the manuscript to see if you can address the criticisms provided. Since Reviewer 2 recommends that the paper be rejected, I may seek additional reviewers for any revised version of the paper.  Therefore, I invite you to submit a revised version of the manuscript that addresses the points raised during the review process, while recognizing that the critical review of Reviewer 2 may still affect a final judgement on the suitability of the paper for publication in PLOS ONE. 

We look forward to receiving your revised manuscript.

Kind regards,

Lee W Cooper, Ph.D.

Section Editor

PLOS ONE

Journal Requirements:

[Research was authorized by SEMARNAT (permit numbers: SGPA/DGVS/04674/10 and SGPA/DGVS/003769/18) and approved by the national park authority National Commission of Protected Natural Areas (CONANP). The article processing fee was supported by COFAA-IPN. SEMARNAT permit authorizations: SGPA/DGVS/04674/10 and SGPA/DGVS/003769/18. We would like to thank the community of Rancho Nuevo Tamaulipas, Mexico and especially Juan Martínez for his support and to Martha López, director of CONANP Tamaulipas.

Institutional Review Board Statement: The study was conducted according to the guidelines of the Mexican authorities, SEMARNAT to study and manage wildlife samples or species.]

 [The author(s) received no specific funding for this work]

Reviewers' comments:

Reviewer's Responses to Questions

**Comments to the Author**

1. Is the manuscript technically sound, and do the data support the conclusions?

Reviewer #1: Yes

Reviewer #2: Partly

2. Has the statistical analysis been performed appropriately and rigorously? 

Reviewer #1: Yes

Reviewer #2: Yes

3. Have the authors made all data underlying the findings in their manuscript fully available?

Reviewer #1: No

Reviewer #2: Yes

4. Is the manuscript presented in an intelligible fashion and written in standard English?

Reviewer #1: Yes

Reviewer #2: Yes

5. Review Comments to the Author

Reviewer #1: Comments about questions above:

Some QA/QC data missing but authors should be able to provide that.

Comments for the Authors Below:

Overall: Good paper, I like that each element analyzed was given context within the paper. Only small details that I would like to be cleared up below,

Introduction:

It is interesting that the deepwater horizon event was mentioned but the current study is not looking at PAHs. It may be good to mention what common heavy metals are found near oil drilling operations. It is a bit confusing to mention it without measuring those contaminants in these turtles.

The deepwater event was also mentioned in the results but I didn’t quite understand why? To explain that a recent event lowered turtle numbers? That the event affected metal accumulation? Some extra wording may be needed to clarify that.

Results:

Something that I notice was missing in the results was the CRM recovery by element. I think its important that there at least be a table showing the recovery of each element in CRM and/or other QA/QC.

I am a bit confused about Table 4. The text mentions a relationship between elements which suggest table 4 lists p-values but then says size and metals are not correlated? The table caption should be specific in saying whether this is R value, R2 , or p-value.

Reviewer #2: This manuscript describes the concentrations of a variety of “standard” trace elements in the blood of Kemp’s Ridley turtles. While topical and perhaps useful to monitoring the health of this critically-endangered species, the use of analytical methods with inadequate detection limits makes the actual use and interpretation of these data questionable. Indeed, most of my review deals with analytical questions, but in their introduction (p. 2, first paragraph) where they say, “Heavy metals are persistent in the environment as they are not degraded over short periods…” Metals are elements, not organic compounds, so they cannot ever “degrade,” they can only change phases (particulate, dissolved, colloidal), locations (e.g., tissue, sediments, etc.), or chemical forms/speciation. Also, the use of the term “heavy metals” is a historic misnaming because some of these are not metals (e.g., As and Se) and many do not have high atomic weights; just call them trace elements, or essential and toxic trace elements. My analytical issues with this paper are because the proper choice of collection and storage, sample treatment (digestions), and sample analyses, all affect the data quality, but were not well described or in fact were not adequate.

Sample collection (p. 2). Are the 10 mL tubes plastic and if so, what type? How long were these samples stored under refrigeration? Long storage can affect concentrations due to loses from adsorption to the container walls, or volatilization (Hg, Se).

Metal analysis (p. 4). What is “fortified with standards of reference?” Was this the standard additions method of calibration? What were the blanks, just deionized water analyzed directly, or where there were process blanks that went through all the handling steps? The recovery percentages between 89-106% are not unusual but a table of recoveries for a statistically rigorous assessment of their method’s accuracy for each element is needed.

Results and discussion. It is not unreasonable to assume that feeding is the main source of trace elements for a higher trophic-level organism (see for example in Luoma and Rainbow’s topical book, Metal Contamination in Aquatic Environments, that covers many of the aspects of this paper). Of their trace elements, Cd, Hg, and Se have probably received the most attention, but the use of an ICP-OES for determining their concentrations is simply useless when their background concentrations are so low to begin with and now you are partitioning into a body fluid. Working near the detection limits means poorer precision and therefore less accuracy. I know that cost may be a factor, but in fact they’re wasting the precious blood and field efforts for simply getting below or near detection limit data.

As is not at all like Cd (p. 8) in its chemistry and biochemistry, not to mention its mode of toxicity. And Hg (pp. 8-9) is a critical contaminant element, so having below detection limits data squanders the interpretation of your data, and comparing your results to others for the same species or other turtles. The topic of bioconcentration can then be adequately addressed using better methods.

6. PLOS authors have the option to publish the peer review history of their article (what does this mean?). If published, this will include your full peer review and any attached files.

Reviewer #1: No

Reviewer #2: No

---

## [Author Response · Author response to Decision Letter 0]

5 Aug 2022

To Editor:

thanks for your support

there was no funding statement for this study. To avoid confusion, changes were made.

---

## [Decision Letter · Decision Letter 1]

30 Aug 2022

Trace elements concentration in blood of nesting Kemp's Ridley turtles (Lepidochelys kempii) at Rancho Nuevo sanctuary, Tamaulipas, Mexico.

PONE-D-22-12917R1

Dear Dr. Ley-Quinonez,

Thank you for making the effort to revise your manuscript, "Trace elements concentration in blood of nesting Kemp's Ridley turtles (Lepidochelys kempii) at Rancho Nuevo sanctuary, Tamaulipas, Mexico." Reviewer #1 has concluded that you have successfully dealt with all of the recommendations that they made, and the manuscript is acceptable for publication in PLOS One.  I did not return the revised manuscript to Reviewer #2, who was in particular critical of the methodolgy used, but thank you for responding to the criticisms made and your efforts to provide more details on the methods. I have carefully reviewed the changes to the manuscript and I judge your revisions to have successfully addressed most of the points made by Reviewer #2. As a result, I am pleased to inform you that your manuscript has been judged scientifically suitable for publication and will be formally accepted for publication once it meets all outstanding technical requirements.

Kind regards,

Lee W Cooper, Ph.D.

Section Editor

PLOS ONE

Additional Editor Comments (optional):

Reviewers' comments:

Reviewer's Responses to Questions

**Comments to the Author**

1. If the authors have adequately addressed your comments raised in a previous round of review and you feel that this manuscript is now acceptable for publication, you may indicate that here to bypass the “Comments to the Author” section, enter your conflict of interest statement in the “Confidential to Editor” section, and submit your "Accept" recommendation.

Reviewer #1: All comments have been addressed

2. Is the manuscript technically sound, and do the data support the conclusions?

Reviewer #1: Yes

3. Has the statistical analysis been performed appropriately and rigorously? 

Reviewer #1: Yes

4. Have the authors made all data underlying the findings in their manuscript fully available?

Reviewer #1: Yes

5. Is the manuscript presented in an intelligible fashion and written in standard English?

Reviewer #1: Yes

6. Review Comments to the Author

Reviewer #1: All my comments were addressed well! There was clarity added to the manuscript in the introduction and methods.

7. PLOS authors have the option to publish the peer review history of their article (what does this mean?). If published, this will include your full peer review and any attached files.

Reviewer #1: No

---

## [Editor Report · Acceptance letter]

7 Oct 2022

PONE-D-22-12917R1 

Trace elements concentration in blood of nesting Kemp’s Ridley turtles (*Lepidochelys kempii*) at Rancho Nuevo sanctuary, Tamaulipas, Mexico. 

Dear Dr. Ley-Quinonez:

I'm pleased to inform you that your manuscript has been deemed suitable for publication in PLOS ONE. Congratulations! Your manuscript is now with our production department. 

Kind regards, 

on behalf of

Dr. Lee W Cooper 

Section Editor

PLOS ONE